# DiTraj: training-free trajectory control for video diffusion transformer

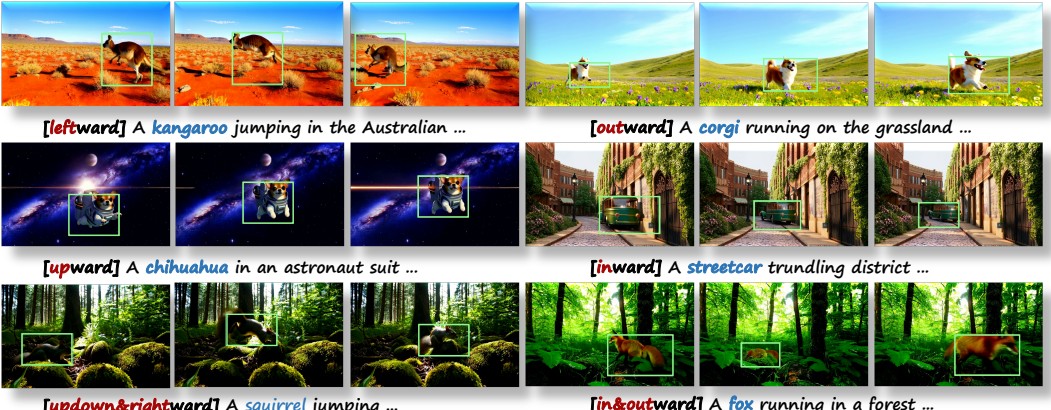

**[leftward]** A *kangaroo* jumping in the Australian ...

**[outward]** A *corgi* running on the grassland ...

**[upward]** A *chihuahua* in an astronaut suit ...

**[inward]** A *streetcar* trundling district ...

**[updown&rightward]** A *squirrel* jumping ...

**[in&outward]** A *fox* running in a forest ...

Figure 1: **Showcase of DiTraj.** We propose DiTraj, a simple but effective training-free framework for trajectory control in text-to-video generation, specifically designed for DiT-based model. Given an input bbox trajectory guidance, DiTraj enables generating high-quality videos that align with the target trajectory.

## ABSTRACT

Diffusion Transformers (DiT)-based video generation models with 3D full attention exhibit strong generative capabilities. Trajectory control represents a user-friendly task in the field of controllable video generation. However, existing methods either require substantial training resources or are specifically designed for U-Net, do not take advantage of the superior performance of DiT. To address these issues, we propose **DiTraj**, a simple but effective training-free framework for trajectory control in text-to-video generation, tailored for DiT. Specifically, first, to inject the object's trajectory, we propose foreground-background separation guidance: we use the Large Language Model (LLM) to convert user-provided prompts into foreground and background prompts, which respectively guide the generation of foreground and background regions in the video. Then, we analyze 3D full attention and explore the tight correlation between inter-token attention scores and position embedding. Based on this, we propose inter-frame Spatial-Temporal Decoupled 3D-RoPE (STD-RoPE). By modifying only foreground tokens' position embedding, STD-RoPE eliminates their cross-frame spatial discrepancies, strengthening cross-frame attention among them and thus enhancing trajectory control. Additionally, we achieve 2.5D-aware trajectory control by regulating the density of position embedding. Extensive experiments demonstrate that our method outperforms previous methods in both video quality and trajectory controllability.

## 1 INTRODUCTION

In recent years, diffusion models have advanced rapidly (Sohl-Dickstein et al., 2015; Ho et al., 2020; Song et al., 2022). Owing to their stable generation process and impressive generation quality, they have gradually become the mainstream for visual generation tasks. Benefiting from large-scale image and video datasets, the architecture of video generation models has evolved from the

Figure 2: Difference in attention mechanisms between U-Net and DiT. Methods based on U-Net fail to achieve sufficient trajectory control and struggle to maintain the consistency of the object's appearance. In contrast, our proposed method enables effective control over the object's trajectory while ensuring the consistency of its appearance.

traditional U-Net (Ronneberger et al., 2015) to the current state-of-the-art Diffusion Transformers (DiT) (Peebles & Xie, 2023). Sora (OpenAI, 2023) has demonstrated that the DiT architecture exhibits excellent scalability and other advantages in video generation tasks, delivering remarkably realistic results. Subsequently, the proposal of numerous DiT-based video generation models—for both open-source (Kong et al., 2025; Wan et al., 2025; Yang et al., 2025b; Zheng et al., 2024) and commercial applications (KlingAI, 2025)—has further advanced the field of video generation.

Researchers not only pursue high-quality generation results but also strive to control the generated video content. Most models offer text-to-video control, in which users guide video generation via prompts to ensure the generated video aligns with the provided textual descriptions. However, relying solely on text often fails to produce the desired results. Although text can control the appearance of objects or scenes, it remains challenging to regulate the trajectory of the object. Controlling the object's position in each frame of a video via its bounding box, thereby governing the object's trajectory, would offer significant convenience for users. To address this task, several methods have been proposed which can be categorized into two types: training-based and training-free approaches. Training-based methods (Zhang et al., 2025; Yang et al., 2024) construct dedicated datasets to train additional modules or directly fine-tune the model's own parameters, but they incur substantial resource costs. In contrast, training-free (Qiu et al., 2024; Jain et al., 2024; Ma et al., 2024; Lian et al., 2024; Chen et al., 2025) methods control object trajectories by modifying noise, constructing attention masks from input cues, assembling noise via inversion and repositioning, or optimizing during inference-time. However, these methods either rely on time-consuming inversion or optimization processes, or are specifically designed for U-Net, failing to leverage the superior performance of DiT. Furthermore, we argue that the U-Net's segregated spatial and temporal attention mechanisms necessitate extensive implicit propagation of visual features, complicating the preservation of consistency for objects undergoing large motions. In contrast, DiT's joint spatial-temporal attention mechanism (i.e., 3D full attention) is more suitable for object trajectory control, as illustrated in Fig. 2. We believe that this inherent mechanism of DiT provides favorable conditions for training-free trajectory control.

In this paper, we propose DiTraj, a training-free framework for trajectory control in text-to-video generation. First, we convert user-provided prompts into foreground and background prompts via rational reasoning using a Large Language Model (LLM); these prompts are then used to guide the generation of foreground and background regions in the video, respectively, by constructing a cross-attention mask between video tokens and prompts. Although the separation guidance enables the control over small movements, it performs poorly for large movements. Through in-depth analysis of the 3D full-attention mechanism, we observe that the attention map exhibits a diagonal highlighting property: tokens with similar position embedding yield higher attention scores. This implies that video tokens tend to pay more attention to tokens with adjacent position embedding either in the spatial or temporal dimension; this phenomenon is also mentioned in previous works (Luo et al., 2025; Wen et al., 2025). This property causes the object in the generated videos to remain relatively static and often confines the object to the overlapping regions of bounding-boxes in the trajectory. To resolve this issue, we propose inter-frame Spatial-Temporal Decoupled 3D-RoPE (STD-RoPE),

a simple but effective method for enhancing attention between foreground tokens across different frames by modifying 3D-RoPE (Su et al., 2023). Specifically, in the layout generation phase of the diffusion process, i.e., the first few steps of the denoising process, we modify the position embedding to align the spatial dimension within the bounding-box of each frame, and preserve the original temporal dimension. The aligned spatial dimension enhance attention between inter-frame foreground tokens, thereby improving control precision; meanwhile, the retained temporal dimension ensures the coherence of the object's motion. However, when we introduce STD-RoPE, some tokens with repeated position embedding emerge, which may lead to the occurrence of artifacts. To address this issue, we introduce a self-attention mask, which eliminates artifacts and further enhances control performance. Additionally, we achieve 2.5D-aware object trajectory control by regulating the density of position embedding in the bounding-box, which is implemented through nearest-neighbor upsampling on the spatial dimension of the position embedding of tokens in the minimum bounding-box. This strategy controls the object's trajectory while simultaneously controlling the distance between the object and the camera. In summary, our contributions are as follows:

- We propose DiTraj, the first training-free framework tailored for DiT for trajectory controllable video generation, which requires no inversion and inference-time optimization. It can be easily adapted to most DiT-based video generation models.

- We introduce foreground-background separation guidance, which injects object trajectory into the video generation process via conditional guidance.

- We propose STD-RoPE: a simple but effective method that improves trajectory control capability by enhancing the attention between foreground tokens across different frames in the layout generation phase of the diffusion process. Furthermore, based on this, we achieve 2.5D-aware object trajectory control by regulating the density of position embedding.

- Extensive experiments demonstrate that DiTraj outperforms existing methods in both video quality and trajectory controllability.

## 2 RELATED WORK

### 2.1 TEXT-TO-VIDEO DIFFUSION MODEL

With the advent of diffusion models (Sohl-Dickstein et al., 2015; Ho et al., 2020; Song et al., 2022), the Text-to-Image (T2I) field has advanced rapidly in recent years, which has further spurred the development of Text-to-Video (T2V) models. Several foundational models (Khachatryan et al., 2023; Blattmann et al., 2023; Guo et al., 2023) have demonstrated robust video generation capabilities by extending T2I model or training on large-scale image and video datasets. Notably, most of these methods adopt the U-Net architecture. Subsequently, the introduction of Sora (OpenAI, 2023) has showcased the scalability and additional advantages of the DiT architecture in video generation. Recent works, such as CogVideoX Yang et al. (2025b), Mochi1 (Genmo, 2024), Wan (Wan et al., 2025), and HunyuanVideo (Kong et al., 2025), have all leveraged the DiT architecture and achieved remarkable performance.

### 2.2 TRAJECTORY CONTROL IN VIDEO GENERATION

As video generation models continue to advance in capability, much research has focused on controlling the trajectories of objects in generated videos. For instance, VideoComposer (Wang et al., 2023) and Control-A-Video (Chen et al., 2024) leverage depth maps, sketches, or motion vectors extracted from reference videos as conditional inputs to control the motion of generated videos. Tora (Zhang et al., 2025) integrates text, visual, and trajectory conditions to generate high-fidelity motion videos. LeviTor (Wang et al., 2025) introduces 3D object trajectory control for image-to-video synthesis, addressing the limitations of 2D drag-based control. However, these methods either require extensive training data and computational resources or demand reference videos for fine-tuning. Meanwhile, several training-free methods have been proposed: Peekaboo (Jain et al., 2024) and Trailblazer (Ma et al., 2024) achieve direct object trajectory control by manipulating the attention mechanism within U-Net; FreeTraj (Qiu et al., 2024) injects trajectories via noise initialization and resampling, alongside proposing a soft mask for enhanced control; Motion-zero (Chen et al., 2025) fuses object trajectories with noise through an inversion process; and LVD (Lian et al., 2024)

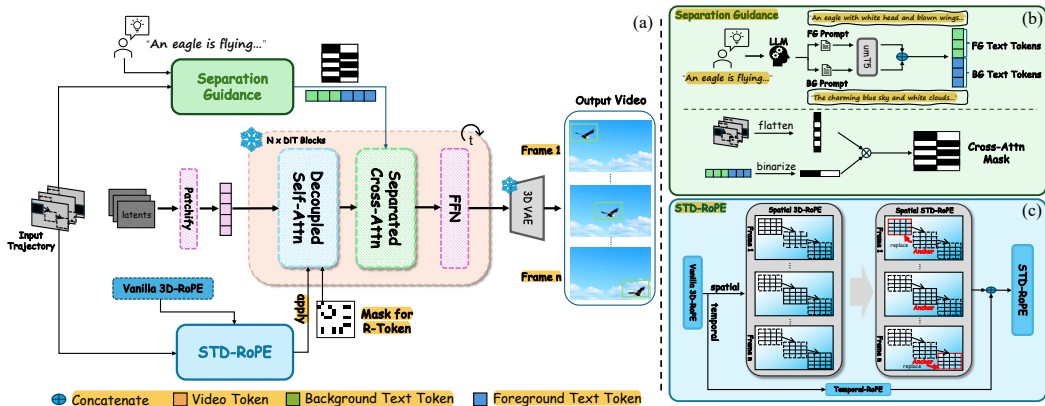

Figure 3: (a) Overview of DiTraj. Given the user-provided prompt and target trajectory, DiTraj achieves training-free trajectory controllable T2V generation. (b) Foreground-background separation conditional guidance. (c) The STD-RoPE processing procedure.

complete trajectory control through inference-time optimization. Constrained by the capabilities of U-Net, the performance of these methods is often unsatisfactory.

## 3 METHOD

In this section, we first briefly introduce 3D full attention (Yang et al., 2025b) and 3D-RoPE (Su et al., 2023)—two key components in video DiT. We then elaborate on DiTraj: first, we present foreground-background separation guidance; next, we describe STD-RoPE for enhancing attention between foreground tokens across different frames, this part begins with an analysis of the attention map, followed by a detailed introduction to STD-RoPE; subsequently, we explain how to addressing tokens with repeated position embedding; finally, we outline our strategy for achieving 2.5D-aware trajectory control. Our method can be extended to most DiT-based models, we use the Wan2.1(Wan et al., 2025) as a concrete example to elaborate on the technical details in this section.

### 3.1 PRELIMINARIES

**3D full attention.** In current video DiT, pixel-level variables $V \in \mathbb{R}^{B \times F \times 3 \times H \times W}$ are first compressed by a 3D-VAE to generate latent variables $z \in \mathbb{R}^{B \times f \times c \times h \times w}$, which are subsequently converted into a sequence of video tokens $x$ with the shape of $(B, L, D)$ via patchifying, where $B$ denotes the batch size, $L = f \times \frac{h}{p} \times \frac{w}{p}$ represents the sequence length, $p$ denotes the patch size, and $D$ indicates the latent dimension. These video tokens are then fed into a transformer block. After position embedding is applied, 3D full attention is computed over the entire token sequence (merged from the three dimensions: height, width, frame). Unlike the spatially and temporally separated attention mechanism in U-Net, 3D full attention enables all tokens across the three dimensions to attend to one another.

**3D-RoPE.** Rotary Position Embedding (RoPE) (Su et al., 2023) is a position embedding method that integrates dependencies on relative positional information into self-attention, it rotates feature vectors in the complex plane, using different rotation angles to represent distinct relative positions. To adapt to video data, 3D-RoPE extends the RoPE: each latent variable in the video tensor is represented by a 3D coordinate (x, y, t), where (x, y) and t correspond to spatial and temporal dimensions, respectively. Then 1D-RoPE is applied independently to each of these three dimensions, and the results are concatenated along the channel dimension to produce the final 3D-RoPE.

### 3.2 FOREGROUND-BACKGROUND SEPARATION GUIDANCE

First, we input the user-provided prompt $\mathcal{P}_{ori}$ with our instruction template into the LLM. Leveraging the LLM's rational reasoning and appropriate semantic expansion, we derive two task-specific

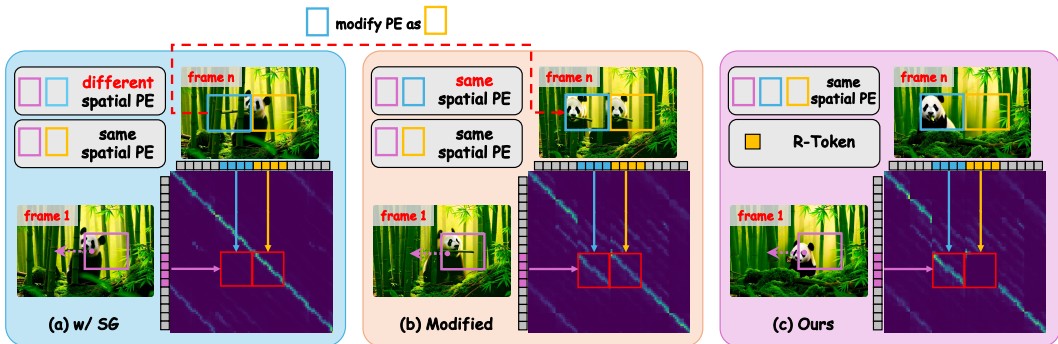

Figure 4: (a) A part of attention map between the first frame and the n-th frame. (b) After modifying the position embedding, regions with the same PE exhibit a similar distribution of attention scores. (c) With STD-RoPE, the attention scores between foreground tokens across different frames are increased in the first step of denoising process. We perform visualization at block 1 in Wan2.1.

prompts: a foreground prompt $\mathcal{P}_{fg}$ (exclusively describing the foreground of the scene in the original prompt) and a background prompt $\mathcal{P}_{bg}$ (exclusively describing the background).

$$\mathcal{P}_{fg}, \mathcal{P}_{bg} = LLM(\mathcal{P}_{ori}) \tag{1}$$

These two prompts serve to guide the generation of the video's foreground and background regions, respectively. Subsequently, we feed two prompts into the text encoder $\mathcal{E}_{text}$ separately, concatenate their output embeddings to form the Union Condition Embedding:

$$C^u = Concatenate(\mathcal{E}_{text}(\mathcal{P}_{fg}), \mathcal{E}_{text}(\mathcal{P}_{bg})), \tag{2}$$

and input this embedding into the cross-attention layer guiding the generation process. To implement foreground-background separation guidance, we construct a cross-attention mask $\mathbf{M}^{cross}$ based on the bounding-box trajectory $\mathbb{T}$, which is composed of $f$-frame bounding-boxes: $\mathbb{T} = \{\mathcal{B}_1, \mathcal{B}_2, \ldots, \mathcal{B}_f\}$. Each bounding-box $\mathcal{B}$ is defined by the relative position coordinates of its top-left and bottom-right corners. Thus, we can determine which video tokens are within the trajectory area.

$$\mathbf{M}_{i,j}^{cross} = \begin{cases} 0, & i \in \mathbb{S}_{fg} \ and \ C_j^u \in \mathcal{E}_{text}(\mathcal{P}_{fg}) \\ 0, & i \notin \mathbb{S}_{fg} \ and \ C_j^u \in \mathcal{E}_{text}(\mathcal{P}_{bg}) \\ -\infty, & other \end{cases} \tag{3}$$

where $\mathbb{S}_{fg} = \{i \mid x_i \in \mathbb{T}\}$. This mask enforces that foreground tokens in the generated video are guided by the foreground prompt, while background tokens are guided by the background prompt. Thus, the cross-attention becomes:

$$CrossAttention(x, C^u, \mathbf{M}^{cross}) = softmax(\frac{(W_q \cdot x) \cdot (W_k \cdot C^u)^T}{\sqrt{D}} + \mathbf{M}^{cross}) \cdot (W_v \cdot C^u) \tag{4}$$

where $W_q$, $W_k$, and $W_v$ are the parameter matrices, which are used to calculate the query, key, and value in the cross-attention, respectively. In this manner, we achieve the injection of the object's trajectory via the foreground-background separation guidance. To achieve better fusion of the foreground and background, we use the separated guidance in the first $t_a$ steps of the entire denoise process and maintain the remaining steps.

### 3.3 STD-RoPE

**Analysis of attention map** After injecting the object's trajectory, the method performs well for small-movement trajectories but fails to achieve precise control for large-movement ones, even if we use separated guidance (SG) throughout the entire denoising process (see Fig. 4(a), where the panda is not within the blue bounding-box in the n-th frame). To investigate this issue, we analyze the attention map between the tokens of the first frame and the n-th frame in the first step of the denoising process. As illustrated in Fig. 4(a), the attention map exhibits distinct diagonal stripes, indicating that tokens at the same spatial position (purple and orange tokens in Fig. 4(a)) have stronger

attention scores, but those in the trajectory (purple and blue tokens in Fig. 4(a)) have weak ones. In other words, tokens with more similar position embedding (PE) tend to yield higher attention scores during self-attention computation. We attribute this phenomenon to the fact that when 3D-RoPE is applied to features, similar 3D-RoPE embeddings lead to comparable rotation angles in the complex plane, resulting in more similar feature representations and thus higher attention scores. To further validate this, we modify the position embedding of the tokens in the bounding-box of the n-th frame (blue tokens in Fig. 4(b)), making its spatial position embedding completely consistent with the bounding-box of the first frame (purple tokens in Fig. 4(b)). The attention map shows that two regions with the same spatial position embedding (blue and orange tokens in Fig. 4(b)) have highly similar attention scores, which results in the two regions being highly similar in the n-th frame. Therefore, we conclude that the poor performance on large-movement trajectories arises from the following issue: the significant spatial span between foreground tokens across different frames leads to their excessively low attention scores. As a result, during the layout generation phase of the denoising process, the latent variables are unable to produce a layout that aligns with the target trajectory.

**STD-RoPE**  To address the aforementioned issue, we propose inter-frame Spatial-Temporal Decoupled 3D-RoPE (STD-RoPE). The algorithm is shown in Alg. 1. This method modifies the position embedding of video tokens to eliminate large spatial discrepancies between foreground tokens across different frames, strengthen their inter-frame attention score, thus ensure the generation of a video spatial layout that conforms to the target trajectory. Specifically, given a bounding-box trajectory $\mathbb{T}$, we can determine which tokens in each frame belong to foreground tokens based on the bounding-box $\mathcal{B}$ in the trajectory. Then we select the spatial dimension of position embedding of foreground tokens of an arbitrary frame as the anchor. We then modify the position embedding of foreground tokens in all other frames to align their spatial di-

---

**Algorithm 1:** STD-RoPE

**Input:** Trajectory $\mathbb{T} = \{\mathcal{B}_1, \mathcal{B}_2, \ldots, \mathcal{B}_f\}$, video tokens $x$
**Output:** STD-RoPE: $PE_{STD}$
1: $PE \leftarrow$ 3D-RoPE$(x)$
2: $PE^{spatial}, PE^{temporal} \leftarrow \text{Split}(PE)$
    ▷Split the $PE$ along the channel dimension
3: $k \leftarrow \text{Random}(1, f)$
4: $anchor \leftarrow PE^{spatial}[k, \mathcal{B}_k]$
    ▷The part of region $\mathcal{B}_k$ in the $k$-th frame of $PE^{spatial}$
5: $i \leftarrow 1$
6: **while** $i \neq f$ **do**
       $PE^{spatial}[i, \mathcal{B}_i] \leftarrow anchor$
       $i \leftarrow i + 1$
      **end**
7: $PE_{STD} \leftarrow Concatenate(PE^{spatial}, PE^{temporal})$
8: **return** $PE_{STD}$

---

mensions with the anchor. This alignment ensures consistent spatial dimension of position embedding for foreground tokens across all frames, eliminating spatial discrepancies and increasing the attention scores between them. Notably, we do not modify the temporal dimension of any token's position embedding, this preserves the coherence and rationality of the object's motion, as well as the continuity and integrity of the entire video. We modify the position embedding in the first $t_b$ steps of the denoising process.

**Mask for R-token**  A critical issue arises after modifying the position embedding: except for the frame corresponding to the anchor, multiple pairs of video tokens with identical position embedding emerge in other frames. This induces a shift in the attention score distribution (similar to the scenario illustrated in Fig. 4(b)), which degrades trajectory control performance and introduces artifacts in generated videos. To address this issue, R-token mask is introduced into the self-attention computation. Specifically, within each frame, tokens with repeated position embedding—excluding foreground tokens—are defined as R-tokens:

$$\mathbb{S}_R = \mathbb{S}_{repeat} - \mathbb{S}_{fg} \tag{5}$$

where $\mathbb{S}_{repeat}$ contains those tokens with repeated position embedding.

The self-attention mask $\mathbf{M}^{self}$ is then constructed to block attention computation between R-Tokens and foreground tokens:

$$\mathbf{M}^{self}_{i,j} = \begin{cases} -\infty, & i \in \mathbb{S}_{fg} \ and \ j \in \mathbb{S}_R \\ -\infty, & i \in \mathbb{S}_R \ and \ j \in \mathbb{S}_{fg} \\ 0, & other \end{cases} \tag{6}$$

This ensures that during self-attention calculation, no two tokens with identical position embedding participate in the attention, thereby mitigating the aforementioned issues.

After applying STD-RoPE and the R-token mask, the attention scores of foreground tokens across different frames are significantly improved during the layout generation phase of the denoising process. Ultimately, a layout that aligns with the target trajectory is generated, as illustrated in Fig. 4(c).

### 3.4 2.5D-AWARE TRAJECTORY CONTROL

To achieve not only control over an object's 2D position in video frames but also regulation of the object's relative distance to the camera (i.e., depth control), we refined the modification of position embedding in STD-RoPE. Specifically, as illustrated in Fig. 5, when a user provides a trajectory with dynamically sized bounding-boxes, we adopt the position embedding of tokens within the smallest bounding-box in the trajectory as the anchor (rather than selecting an arbitrary frame in sec 3.3). For all other frames, we modify the position embedding of tokens within their

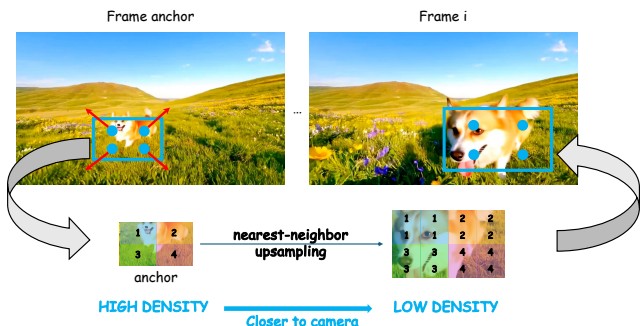

Figure 5: 2.5D-aware trajectory control by nearest-neighbor upsampling from the anchor.

respective bounding-boxes such that their spatial dimensions align with those of the anchor, where the anchor's position embedding is first upsized to match the size of the target frame's bounding-box via nearest-neighbor upsampling. Thus, in the layout generation process, we use the density of position embedding values to control the distance between objects and the camera. This design allows users to implement 2.5D-aware trajectory control by defining a bounding-box trajectory with dynamic sizes, where variations in bounding-box size correspond to changes in the object's depth relative to the camera. The examples are shown in the right side of Fig. 1.

## 4 EXPERIMENT

### 4.1 SETTINGS

To validate the generalizability of our method, we adopt two DiT-based models—Wan2.1 (Wan et al., 2025) and CogVideoX (Yang et al., 2025b)—as our pre-trained model. We set the number of inference steps to 50, with $t_a$ set to 30 and $t_b$ set to 5. Additional details are provided in the supplementary material.

We compared two categories of methods: training-free methods and training-/optimizing-based methods. The training-free methods include Peekaboo (Jain et al., 2024), Trailblazer (Ma et al., 2024), and FreeTraj (Qiu et al., 2024); the training-/optimizing-based methods include Tora (Zhang et al., 2025), Direct-a-video (Yang et al., 2024), and LVD (Lian et al., 2024).

### 4.2 QUALITATIVE COMPARISON

As shown in Fig. 6, our method achieves the best performance in both control capability and object consistency maintenance, outperforming all other methods. Peekaboo (Jain et al., 2024), FreeTraj (Qiu et al., 2024), and direct-a-video (Yang et al., 2024) exhibit poor control capability, with the target object in the generated videos failing to align with the target trajectories. Although Trailblazer (Ma et al., 2024) and LVD (Lian et al., 2024) realize trajectory control, their subjects are damaged, which seriously impairs the quality of the generated videos. Based on CogvideoX (Yang et al., 2025b), our method generates videos of higher quality than Tora (Zhang et al., 2025).

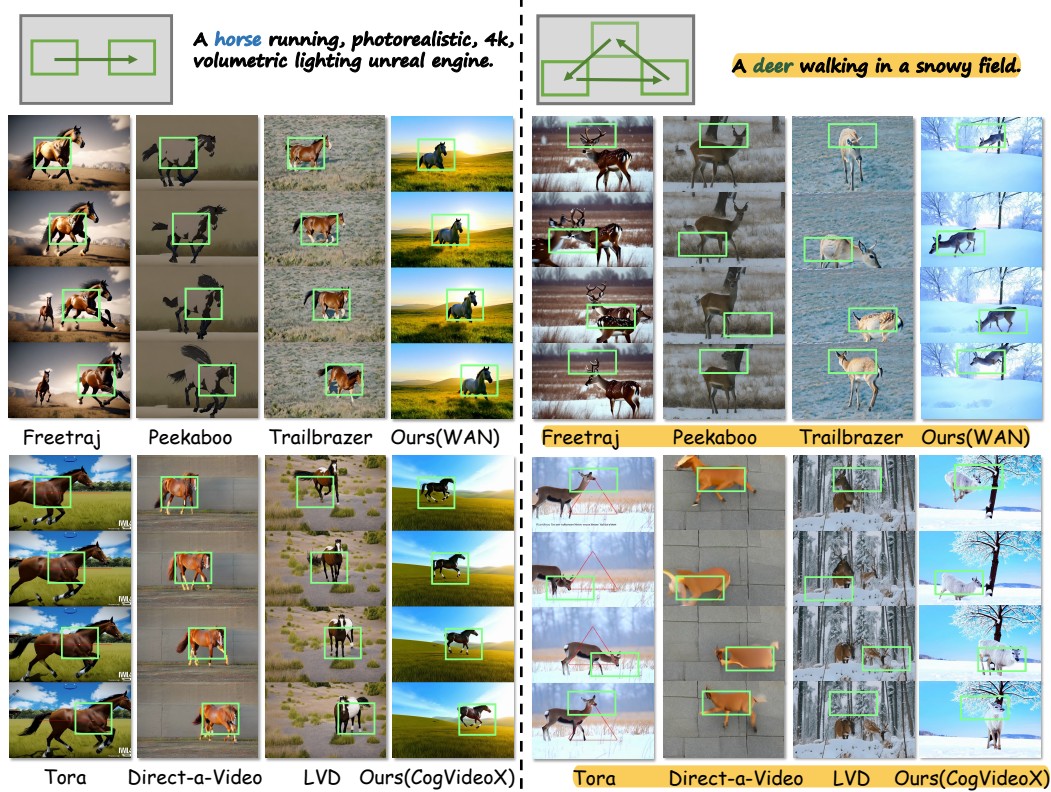

Figure 6: Qualitative comparison with state-of-the-art methods.

Table 1: **Comparison with state-of-the-art methods**. **Red** and **Blue** denote the best and second best results, respectively.

| Method | Video Quality | | | | | Trajectory Control | | | |
|---|---|---|---|---|---|---|---|---|---|
| | SC↑ | BC↑ | MS↑ | AQ↑ | IQ↑ | Cov↑ | mIoU↑ | CD↓ | AP50↑ |
| **Training-/Optimizing-Based Methods** | | | | | | | | | |
| Tora (Zhang et al., 2025) | **0.936** | 0.956 | 0.988 | 0.541 | 0.640 | **0.95** | 21.3 | 0.17 | 3.4 |
| Direct-a-Video (Yang et al., 2024) | 0.923 | 0.931 | 0.959 | 0.478 | 0.551 | 0.83 | 37.7 | 0.14 | 22.1 |
| LVD (Lian et al., 2024) | 0.931 | 0.925 | 0.974 | 0.593 | 0.642 | 0.85 | 36.6 | 0.15 | 20.7 |
| **Training-Free Methods** | | | | | | | | | |
| Peekaboo (Jain et al., 2024) | 0.920 | 0.943 | 0.986 | 0.482 | 0.544 | 0.84 | 34.0 | 0.17 | 18.7 |
| TrailBlazer (Ma et al., 2024) | 0.925 | 0.949 | 0.971 | 0.537 | **0.671** | 0.86 | 40.8 | 0.15 | 49.1 |
| FreeTraj (Qiu et al., 2024) | 0.935 | 0.950 | 0.968 | **0.584** | 0.650 | 0.94 | 37.2 | **0.11** | 26.3 |
| Ours (CogvideoX) | 0.935 | **0.956** | **0.990** | 0.580 | 0.652 | 0.94 | **45.2** | 0.14 | **58.8** |
| Ours (Wan2.1) | **0.937** | **0.957** | **0.990** | **0.627** | **0.677** | **0.96** | **47.3** | **0.09** | **50.5** |

## 4.3 QUANTITATIVE COMPARISON

**Evaluation metrics** To evaluate video quality, we report five dimensions in VBench (Huang et al., 2023): Subject Consistency (SC), Background Consistency (BC), Motion Smoothness (MS), Aesthetic Quality (AQ) and Imaging Quality (IQ). For trajectory control performance, we follow the evaluation protocol proposed in (Jain et al., 2024): first, we use the off-the-shelf object detection model OWL-ViT-large (Minderer et al., 2022) to extract bounding-boxes of target objects in the generated videos; subsequently, we compute four metrics to quantify control accuracy: Coverage (Cov), mean Intersection over Union (mIoU), Center Distance (CD), and Average Precision at 50% IoU (AP50). Here, Cov and CD represent the fraction of generated videos that the bboxes detected

Table 2: User study. **Red** denotes the best results.

| Method | Tora | DAV | LVD | Peekaboo | TrailBlazer | FreeTraj | Ours |
|---|---|---|---|---|---|---|---|
| Trajectory Alignment | 9.72% | 5.12% | 4.56% | 1.93% | 12.89% | 3.90% | **61.88%** |
| Video-Text Alignment | 13.60% | 3.24% | 10.35% | 2.24% | 4.77% | 6.73% | **59.07%** |
| Video Quality | 11.28% | 3.30% | 7.71% | 6.48% | 3.96% | 4.17% | **63.10%** |

in more than half of the frames and the distance between the centroid of the generated subject and input mask, respectively.

As illustrated in Table 1, compared with those U-Net-based training-free methods, our approach based on Wan2.1 outperforms all other methods across the five dimensions of video quality. And it significantly surpasses other methods in the four dimensions related to trajectory control, with improvements of 2.1%, 15.9%, 18.2%, and 2.9% respectively over the second-ranked method in terms of Cov, mIoU, CD, and AP50. Compared with those training/optimizing-based methods, our approach also achieves the best performance across all metrics.

In addition, a user study is employed for the assessment of human preferences. 24 participants are instructed to select the best video in three evaluation aspects: trajectory alignment, video-text alignment, and video quality. As shown in Table 2, DiTraj outperforms the baseline methods by a significant margin, confirming the superiority of our approach in terms of trajectory alignment, video-text alignment, and video quality.

## 4.4 ABLATION STUDY

To validate the effectiveness of foreground-background separation guidance (SG) and STD-RoPE, we conducted experiments with Wan2.1 in three test settings: the original model, the model with only separation guidance (SG) and the complete DiTraj. As shown in Table 3, compared with the original model, the model with SG achieves improvements in both video quality and trajectory control capability (except in Cov dimension). Compared with the model using SG, the full DiTraj framework shows a slight decrease of 0.4%, 0.1%, and 2.0% in the three video quality metrics (SC, MS, IQ), respectively; however, it delivers substantial improvements of 33.6%, 25.0%, and 97.3% in the three trajectory control metrics (mIoU, CD, AP50). As illustrated in Fig. 7, compared with original model, the integration of SG yields notable alterations in the video layout; however, the object trajectory exhibits insufficient consistency with the target trajectory (part of the horse's body extends beyond the bounding box range). In contrast, following the introduction

Table 3: **Ablation study**. **Red** denotes the best results.

| Method | Video Quality | | | Trajectory Control | | | |
|---|---|---|---|---|---|---|---|
| | SC↑ | MS↑ | IQ↑ | Cov↑ | mIoU↑ | CD↓ | AP50↑ |
| original | 0.924 | 0.976 | 0.608 | **0.97** | 23.7 | 0.17 | 7.7 |
| w/ SG | **0.941** | **0.991** | **0.691** | 0.96 | 35.4 | 0.12 | 25.6 |
| DiTraj | 0.937 | 0.990 | 0.677 | 0.96 | **47.3** | **0.09** | **50.5** |

A horse running, photorealistic, 4k, volumetric lighting unreal engine.

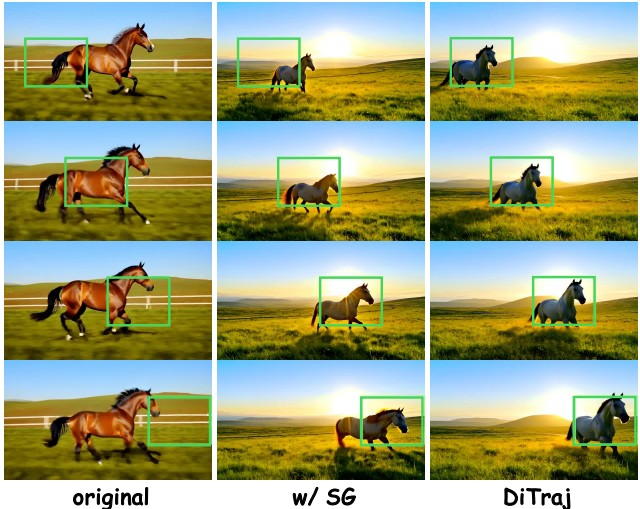

original      w/ SG      DiTraj

Figure 7: **Ablation study about proposed modules**. We gradually incorporate the modules we proposed into the base model to verify their effectiveness.

of STD-RoPE, the object trajectory achieves complete alignment with the target trajectory, enabling more precise trajectory control.

## 5 CONCLUSION

We present DiTraj, the first DiT-specific training-free method for object trajectory control in T2V generation, without inversion and inference-time optimization. Firstly, we inject the object trajectory into the generation process by foreground-background separation guidance. Subsequently, we propose STD-RoPE to eliminate the spatial dimension discrepancy between foreground tokens across different frames, increasing the attention score among them during the layout generation phase of the denoising process, thereby enhancing the trajectory control capability. Moreover, we achieve 2.5D-aware trajectory control by regulating the density of position embedding. We reveal the potential connection between position embedding and attention score, and use it to control the generation of video layouts. We hope that our work can offer valuable insight for future work on DiT-based controllable trajectory video generation.

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

## A Appendix

### A.1 Use of Large Language Models in paper writing

In the process of writing our article, we used large language models (LLMs) to aid and polish writing. Specifically, we leverage LLMs to check for grammatical errors and correct punctuation usage. Additionally, we utilize LLMs to enhance the fluency of some sentences and the accuracy of word choice in the paper, thereby improving its readability. No LLMs are employed to generate new ideas, and the research process is conducted by the authors.

### A.2 Implementation

#### A.2.1 Hyperparameters

We use Qwen3 (Yang et al., 2025a) as our LLM. For Wan-based (Wan et al., 2025) DiTraj, the inference resolution is fixed at 480×832 pixels and the video length is 81 frames, the scale of the classifier-free guidance is set to 5. For CogvideoX-based (Yang et al., 2025b) DiTraj, the inference resolution is fixed at 480×720 pixels and the video length is 49 frames, the scale of the classifier-free guidance is set to 6. All experiments are conducted on a single NVIDIA A100 GPU.

For quantitative comparison, we generate a total of 560 videos for each inference method, utilizing 56 prompts. We initialize 10 random initial noises for each prompt for direct inference.

It is worth noting that for the evaluation of trajectory control capability, regarding all bounding-box-based trajectory control methods (Jain et al., 2024; Ma et al., 2024; Qiu et al., 2024; Lian et al., 2024; Yang et al., 2024) (i.e., all methods except Tora (Zhang et al., 2025)), we use the bounding-box trajectory corresponding to each prompt as the condition to guide generation; whereas for Tora, which adopts a point-based trajectory guidance condition, we use the center point of the bounding-box corresponding to each prompt as the condition for guiding generation. For all these methods, we followed their original models and parameter settings as reported in their respective research papers.

#### A.2.2 Prompts

Our prompt set is mostly extended from previous baselines (Jain et al., 2024; Ma et al., 2024), and we manually designed a bounding-box trajectory for each prompt to ensure the diversity and rationality. The prompt word(s) in bold case is the subject for positioning:

- A **woodpecker** climbing up a tree trunk.
- A **squirrel** descending a tree after gathering nuts.
- A **bird** diving towards the water to catch fish.
- A **frog** leaping up to catch a fly.
- A **parrot** flying upwards towards the treetops.
- A **squirrel** jumping from one tree to another.
- A **rabbit** burrowing downwards into its warren.
- A **satellite** orbiting Earth in outer space.
- A **skateboarder** performing tricks at a skate park.
- A **leaf** falling gently from a tree.
- A **paper plane** gliding in the air.
- A **bear** climbing down a tree after spotting a threat.
- A **duck** diving underwater in search of food.
- A **kangaroo** hopping down a gentle slope.
- An **owl** swooping down on its prey during the night.
- A **hot air balloon** drifting across a clear sky.
- A **red double-decker bus** moving through London streets.

- A **jet plane** flying high in the sky.
- A **helicopter** hovering above a cityscape.
- A **roller coaster** looping in an amusement park.
- A **streetcar** trundling down tracks in a historic district.
- A **rocket** launching into space from a launchpad.
- A **deer** walking in a snowy field.
- A **horse** grazing in a meadow.
- A **fox** running in a forest clearing.
- A **swan** floating gracefully on a lake.
- A **panda** walking and munching bamboo in a bamboo forest.
- A **penguin** walking on an iceberg.
- A **lion** walking in the savanna grass.
- An **owl** flying in a tree at night.
- A **dolphin** just breaking the ocean surface.
- A **camel** walking in a desert landscape.
- A **kangaroo** jumping in the Australian outback.
- A **colorful hot air balloon** tethered to the ground.
- A **corgi** running on the grassland on the grassland.
- A **corgi** running on the grassland in the snow.
- A **man** in gray clothes running in the summer.
- A **knight** riding a horse on a race course.
- A **horse** galloping on a street.
- A **lion** running on the grasslands.
- A **dog** running across the garden, photorealistic, 4k.
- A **tiger** walking in the forest, photorealistic, 4k, high definition.
- **Iron Man** surfing on the sea.
- A **tiger** running in the forest, photorealistic, 4k, high definition.
- A **horse** running, photorealistic, 4k, volumetric lighting unreal engine.
- A **panda** surfing in the universe.
- A **chihuahua** in an astronaut suit floating in the universe, cinematic lighting, glow effect.
- An **astronaut** waving his hands on the moon.
- A **horse** galloping through a meadow.
- A **bear** running in the ruins, photorealistic, 4k, high definition.
- A **barrel** floating in a river.
- A **dark knight** riding a horse on the grassland.
- A **wooden boat** moving on the sea.
- A **red car** turning around on a countryside road, photorealistic, 4k.
- A **majestic eagle** soaring high above the treetops, surveying its territory.
- A **bald eagle** flying in the blue sky.

### A.2.3 INSTRUCTION TEMPLATE FOR FOREGROUND-BACKGROUND SEPARATION GUIDANCE

The instruction template input into the LLM in Sec 3.2 is as follows:

*You are a prompt engineer. Users will provide you with a prompt for generating videos. Your task is to understand this prompt, distinguish the main subject (foreground) and the background, and finally return a prompt that only describes the main subject and a prompt that only describes the background. The requirements are as follows: 1. The output format is: foreground_prompt: [prompt describing only the main subject] background_prompt: [prompt describing only the background] 2. The lengths of foreground_prompt and background_prompt should be around 80-100 words long. 3. The foreground_prompt should include a description of a close-up shot, indicating that the main subject fills the entire frame. 4. The content described in the background_prompt should be consistent with the background content of the prompt provided by the user, and it must not contain fields related to the main subject, nor include information about the foreground subject. Example: User: Realistic photography style, a medium-sized gray-and-white dog with fluffy fur running to the right. The dog has bright black eyes, perked ears, and a wagging tail. Its legs are in mid-stride, paws lifting off the ground, mouth slightly open as if panting. The background is a sunlit green lawn with a few scattered flowers. The camera follows the dog in a smooth tracking shot, capturing its energetic movement. Medium shot from a low angle, emphasizing the dog's speed and vitality. foreground_prompt: Realistic photography style, a medium-sized gray-and-white dog with fluffy fur running to the right. The dog has bright black eyes, perked ears, and a wagging tail. Its legs are in mid-stride, paws lifting off the ground, mouth slightly open as if panting. The camera follows the dog in a smooth tracking shot, capturing its energetic movement. Close shot from a low angle, emphasizing the dog's speed and vitality. background_prompt: Hyper-realistic photography, a lush garden bathed in soft afternoon sunlight. Vibrant roses in red, pink, and yellow bloom densely on climbing trellises, while green ivy creeps up weathered stone walls. A small stone fountain gurgles gently in the center, with water rippling and reflecting the sky. Butterflies flit between lavender bushes, and a honeybee hovers above a daisy. The grass is neatly trimmed, with a winding gravel path. I will now provide the prompt for you. Please directly output the foreground_prompt and background_prompt follow the format without extra responses and quotation mark.*

### A.3 MORE EXPERIMENT

#### A.3.1 INFERENCE OVERHEAD

We also evaluated the additional inference overhead incurred by DiTraj. As shown in Table 4, DiTraj results in an extra inference time of 5.9% and 4.7% on Wan2.1-1.3B and CogvideX-5B, respectively. Our method achieves high-quality trajectory control with a low additional inference overhead.

Table 4: Inference overhead.

| Method | Inference time(s) |
|---|---|
| Wan2.1-1.3B | 185 |
| DiTraj (Wan2.1-1.3B) | $196_{\uparrow\,5.9\%}$ |
| CogvideoX-5B | 213 |
| DiTraj (CogvideoX-5B) | $223_{\uparrow\,4.7\%}$ |

#### A.3.2 ABLATION STUDY OF $t_a$ AND $t_b$

Regarding the selection of $t_a$ and $t_b$, we conducted ablation experiments on them based on Wan2.1 respectively. For $t_a$, as illustrated in Fig. 8 and Table 5, when $t_a$ is greater than 5, the generated videos and their quantitative results are very close. For $t_b$, as illustrated in Fig. 9, excessively small values (e.g. 1) will lead to insufficient control ability, while excessively large values (e.g. 10, 20) will result in the appearance of artifacts. Therefore, we selected 30 and 5 as the relatively optimal values for $t_a$ and $t_b$, respectively.

Table 5: Ablation study on $t_a$ and $t_b$. **Bold** denote the best results.

| $t_a$ | $t_b$ | Video Quality | | | Trajectory Control | | | |
|---|---|---|---|---|---|---|---|---|
| | | SC↑ | MS↑ | IQ↑ | Cov↑ | mIoU↑ | CD↓ | AP50↑ |
| 0 | 0 | 0.924 | 0.976 | 0.608 | **0.97** | 23.7 | 0.17 | 7.7 |
| 5 | 0 | 0.934 | 0.982 | 0.687 | 0.97 | 32.1 | 0.15 | 17.9 |
| 30 | 0 | **0.941** | **0.991** | **0.691** | 0.96 | 35.4 | 0.12 | 25.6 |
| 50 | 0 | 0.939 | 0.986 | 0.688 | 0.95 | 36.6 | 0.12 | 25.9 |
| 30 | 1 | 0.939 | 0.991 | 0.688 | 0.96 | 37.9 | 0.11 | 30.7 |
| 30 | 5 | 0.937 | 0.990 | 0.677 | 0.96 | **47.3** | **0.09** | **50.5** |
| 30 | 10 | 0.928 | 0.972 | 0.642 | 0.95 | 45.1 | 0.11 | 47.4 |
| 30 | 20 | 0.911 | 0.964 | 0.621 | 0.92 | 41.1 | 0.12 | 44.7 |

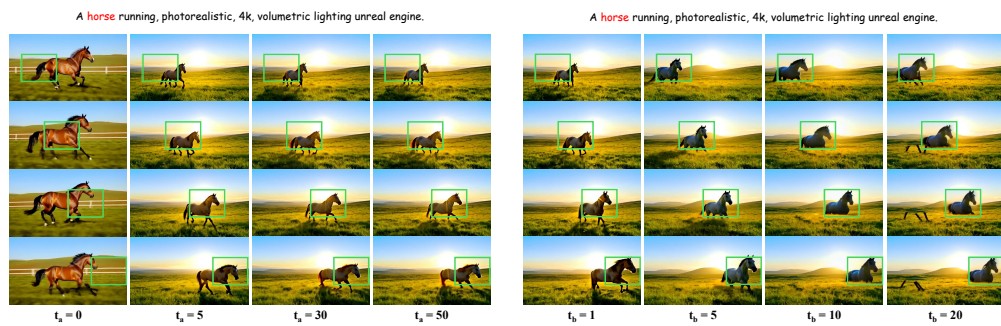

Figure 8: Results generated by varying $t_a$ when $t_b$ is fixed to 0.

Figure 9: Results generated by varying $t_b$ when $t_a$ is fixed to 30.

Table 7: Experiments on two different ways of handling the temporal dimension. **Red** denotes the best results.

| Method | Video Quality | | | | | Trajectory Control | | | |
|---|---|---|---|---|---|---|---|---|---|
| | SC↑ | BC↑ | MS↑ | AQ↑ | IQ↑ | Cov↑ | mIoU↑ | CD↓ | AP50↑ |
| Retaining | **0.937** | **0.957** | **0.990** | **0.627** | **0.677** | **0.96** | 47.3 | **0.09** | 50.5 |
| Aligning | 0.930 | 0.952 | 0.957 | 0.619 | 0.676 | 0.96 | **47.7** | 0.10 | **50.8** |

### A.3.3 ADDITIONAL METRICS ON VIDEO QUALITY

We have supplemented the FVD, FID, and IS metrics. We used 500 randomly selected videos from AnimalKingdom (Ng et al., 2022) as the real distribution. The results are shown in Table 6.

### A.3.4 TEMPORAL DIMENSION OF STD-ROPE

In STD-RoPE, the purpose of retaining the temporal dimension is to maintain the coherence of motion in the foreground region of each frame. If we align the temporal dimensions of all frames, the position embedding of the foreground region in each frame will be identical. This will cause the content in the foreground region of all frames to become almost the same, leading to rigid and unsmooth object motions. To verify this, we conducted an experiment comparing the two schemes of retaining the temporal dimension and aligning the temporal dimension. The Table 7 indicates that aligning the temporal dimension leads to a decline in video quality, particularly in terms of MS (Motion Smoothness).

Table 6: Additional metrics on video quality. **Bold** denote the best results.

| Method | FVD↓ | FID↓ | IS↑ |
|---|---|---|---|
| FreeTraj (Qiu et al., 2024) | 1946 | 101.4 | 14.78 |
| Peekaboo (Jain et al., 2024) | 1287 | 90.64 | 13.08 |
| Trailblazer (Ma et al., 2024) | 1336 | 89.32 | 15.25 |
| Direct-a-Video (Yang et al., 2024) | 1455 | 102.8 | 13.73 |
| LVD (Lian et al., 2024) | 1288 | 99.81 | 14.19 |
| Tora (Zhang et al., 2025) | 1198 | 91.33 | 15.79 |
| **DiTraj** | **1168** | **89.08** | **15.91** |

### A.4 MORE RESULTS

More results are shown in Fig. 10, Fig. 11 and Fig. 12.

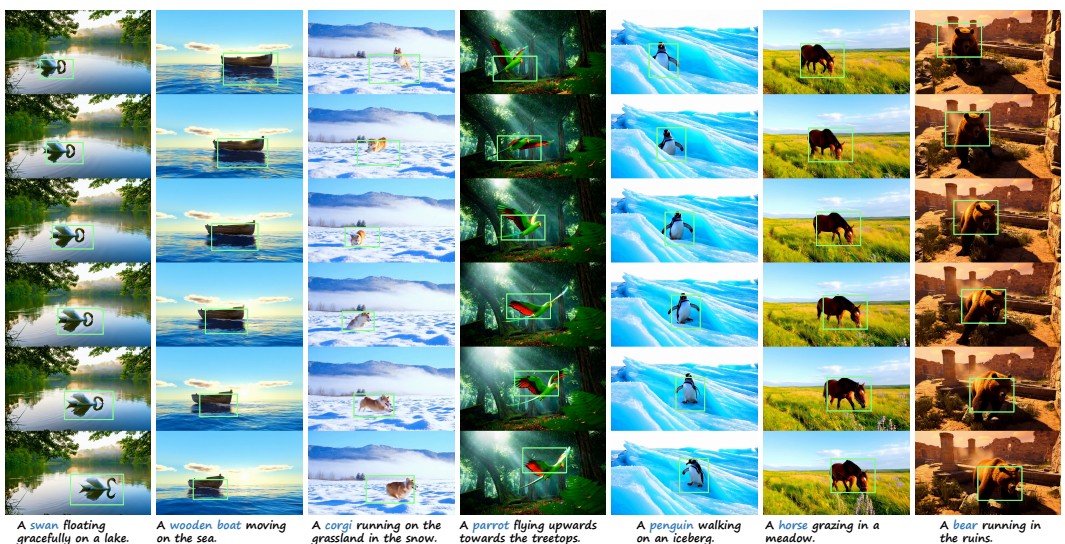

Figure 10: More results generated from DiTraj.

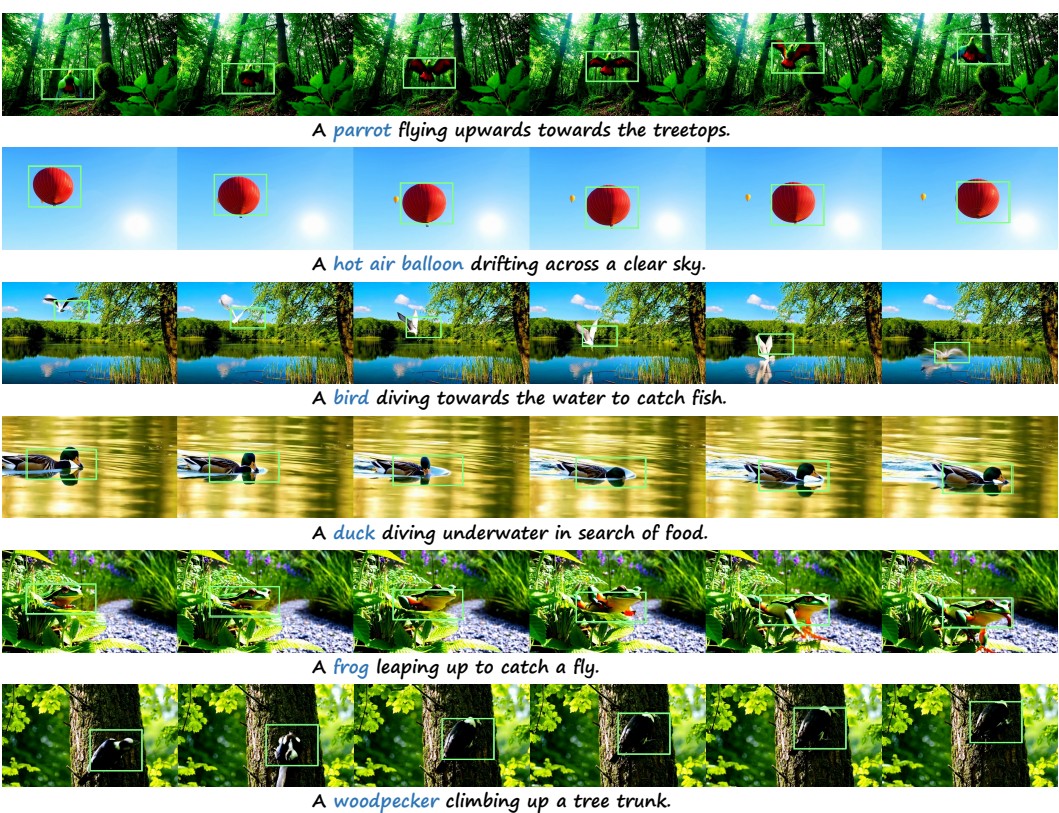

Figure 11: More results generated from DiTraj.

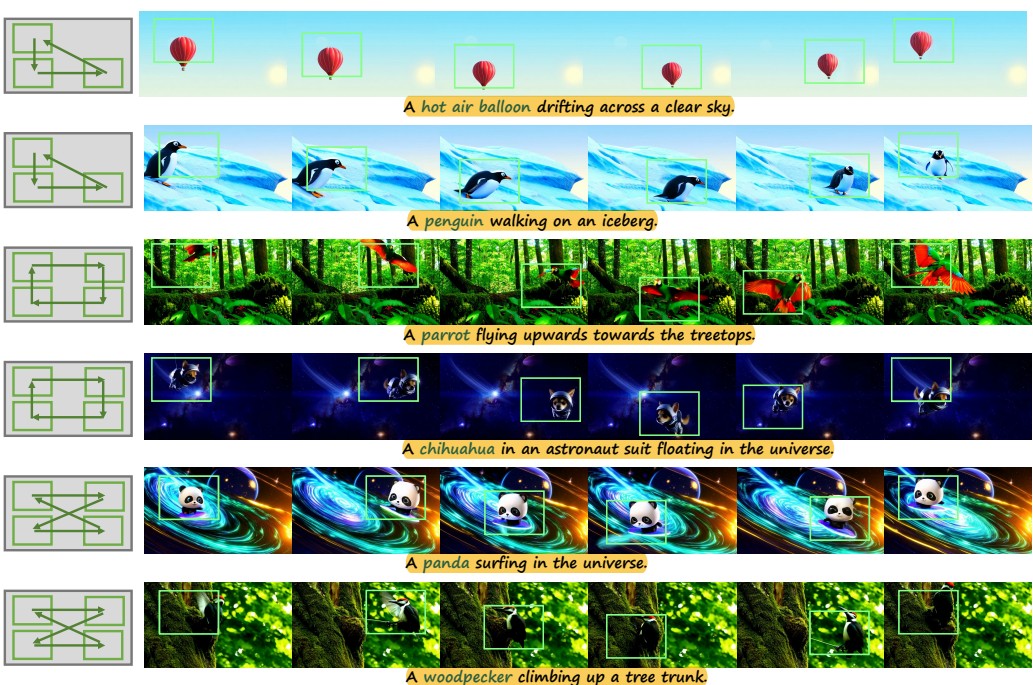

Figure 12: More results generated from DiTraj.

