# OpenReview forum: "DiTraj: Training-free Trajectory Control For Video Diffusion Transformer"
_ICLR.cc/2026/Conference — Submitted to ICLR 2026_

### Official Review · Reviewer_CB3N · 2025-10-21

**Soundness:** 3
**Presentation:** 3
**Contribution:** 3
**Rating:** 6
**Confidence:** 4

**Summary:**

This paper presents DiTraj, a training-free framework for trajectory control in text-to-video generation tailored for DiT models. The method introduces two key innovations: foreground-background separation guidance via LLM prompt decomposition and inter-frame Spatial-Temporal Decoupled 3D-RoPE. STD-RoPE modifies position embeddings to enhance cross-frame attention among foreground tokens, improving trajectory alignment for large movements. Additionally, 3D-aware control is achieved by regulating position embedding density. Experiments on DiT-based models demonstrate superior performance in video quality and trajectory controllability compared to training-free and training-based baselines.

**Strengths:**

(1) DiTraj fills a critical gap in training-free trajectory control for DiT-based video models, eliminating the need for expensive fine-tuning or inference-time optimization. Its compatibility with existing DiT architectures enhances real-world applicability.

(2) Quantitative results and user studies consistently show DiTraj outperforms baselines in both trajectory alignment and video quality. The ablation study validates the contributions of foreground-background separation and STD-RoPE.

(3) The analysis of 3D full attention mechanisms and the diagnosis of poor large-movement control as a result of low cross-frame attention scores demonstrate technical depth. STD-RoPE’s design to decouple spatial and temporal dimensions addresses this issue logically.

(4) Readable Presentation: The manuscript is well-structured, with clear explanations of methodologies and intuitive visualizations.

**Weaknesses:**

(1) The core idea of injecting control via cross-attention masks (foreground-background separation) follows standard practices in conditional diffusion models. There are no obvious bottlenecks or difficulties in extending it to the field of trajectory controled video generation.

(2) Ambiguity in STD-RoPE Mechanism: While STD-RoPE improves performance, modifying position embeddings to eliminate spatial discrepancies raises concerns: the altered embeddings no longer encode true spatial positions, yet the paper does not clarify their semantic meaning or address potential side effects.
Underjustified Design Choices: The decision to retain temporal dimensions while aligning spatial embeddings is empirically effective but lacks theoretical justification. Why does preserving temporal embedding alone ensure motion coherence?

**Questions:**

(1) The foreground-background separation guidance relies on LLM prompt decomposition. How robust is this process to ambiguous user prompts (e.g., prompts with multiple foreground objects or vague scene descriptions)?

(2) STD-RoPE modifies position embeddings to "align" spatial dimensions on different positions across different frames, making these embeddings no longer represent actual spatial coordinates. What do the modified embeddings encode, and how do they avoid confusing the model’s understanding of scene layout?

---

> ### Author Response · Authors · 2025-11-21
>
> We sincerely thank the reviewer for the insightful comments.
>
> ---
>
> > **W1: The core idea of injecting control via cross-attention masks follows standard practices in conditional diffusion models. There are no obvious bottlenecks or difficulties in extending it to the field of trajectory controled video generation.**
>
> Indeed, the technique of injecting different conditions through cross-attention masks has indeed been widely explored in conditional diffusion models, especially in the realm of image generation. Although the direct application of this technique to trajectory-controllable video generation tasks may seem straightforward, there are many issues involved. Our proposed DiTraj represents the first training-free framework specifically tailored for DiT, focusing on trajectory-controllable video generation. Our core contribution lies in addressing the specific challenges encountered when adapting this technique to video generation, thus ensuring its effectiveness in this domain. Subsequent to validating its superiority, we believe that our work provides valuable insights for future video generation research utilizing this technique.
>
>
> ---
>
> > **W2: Ambiguity in STD-RoPE Mechanism(Q2) and Underjustified Design Choices**
>
> - **Ambiguity in STD-RoPE Mechanism(Q2)**
>
>     STD-RoPE aligns the spatial dimensions of foreground region tokens' position embedding across different frames. Intuitively, this means that all foreground regions are encoded within the same area of the video, thereby augmenting the inter-frame attention interaction among foreground tokens (theoretically, in RoPE, tokens with closer position have higher feature similarity, consequently yielding higher attention scores). For details, we assume that the k-th frame is selected. We can identify the anchor-tokens based on $\mathcal{B}_k$ and get their position embedding $PE[k,\mathcal{B}\_k]$, i.e., the anchor. Subsequently, we align the spatial dimensions of the position embedding of the foreground region tokens in all other frames with it:
>     $$
>     PE\_{spatial}[i, \mathcal{B}\_i] ← PE\_{spatial}[k, \mathcal{B}\_k],i \in \{1,...,k-1,k+1,...,f\}
>     $$
>
>     We apply STD-RoPE only in the first few steps ($t_b$) of the entire inference process, aiming to modify the layout of the generated video. In all subsequent steps, we use the original position embedding to maintain the spatial positions of all tokens. Regarding your concerns, and therefore, we have conducted ablation experiments on $t_b$ in the **Appendix A.3.2 (Table 5 and Figure 9)**. The experimental results show that an excessively large $t_b$ will lead to artifacts and misalignment, whereas an appropriate $t_b$ enables trajectory control while maintaining video quality.
>
> - **Underjustified Design Choices**
>
>     The purpose of retaining the temporal dimension is to maintain the motion coherence of the foreground region in each frame. If the temporal dimensions of all frames are aligned, the position embedding of the foreground region in each frame will be identical. This will lead to nearly uniform content in the foreground region across all frames, thereby resulting in rigid and unsmooth object movements.
>
>     To verify this, we have added a comparative experiment between two schemes: retaining the temporal dimension and aligning the temporal dimension. As indicated in below table, aligning the temporal dimension leads to a decline in video quality, with a particularly significant drop in terms of MS (Motion Smoothness) metric.
>
>     ||SC↑|BC↑|MS↑|AQ↑|IQ↑|Cov↑|mIoU↑|CD↓|AP50↑|
>     |:-----|:------|:------|:------|:------|:------|:------|:------|:------|:------|
>     |retaining|**0.937**|**0.957**|**0.990**|**0.627**|**0.677**|**0.96**|47.3|**0.09**|50.5|
>     |aligning|0.930|0.952|0.957|0.619|0.676|0.96|**47.7**|0.10|**50.8**|
>
>     Thank you for your reply. We will include this part in the revised version of the paper.
>
> ---
> > **Q1: how robust is the LLM for foreground-background separation**
>
> Thank you for your feedback. To verify the robustness of the LLM, we collect 120 prompts for foreground-background separation, which cover complex scenarios and ambiguous expressions (e.g., "A Red Fox Navigating the Winter Wetland Maze.", "A bird on a car." etc.). These prompts are combined with our template and input into the LLM to obtain the results, which are then evaluated by humans to determine whether they meet expectations.
>
> |model| success | error | acc |
> |:-----|:------|:------|:------|
> |GPT-4o| 119 | 1 | 99.17%|
> |Qwen3-7B| 117 | 3 | 97.50%|
>
> The experimental results demonstrate that the LLM rarely makes mistakes and exhibits high reliability.
>
> ---
> We are grateful again for your insightful comments. Please let us know if any further elaboration is needed.

---

> ### Author Response · Authors · 2025-11-27
>
> Dear Reviewer CB3N,
>
> We would like to kindly check if our previous response clarifies your questions. We are happy to provide any further details if needed.
>
> Thank you for your time and effort in reviewing our paper.
>
> Best regards,
>
> The Authors

---

### Official Review · Reviewer_ddfm · 2025-10-27

**Soundness:** 3
**Presentation:** 2
**Contribution:** 3
**Rating:** 4
**Confidence:** 4

**Summary:**

The paper proposes a training-free method for trajectory control in video diffusion models. This was achieved by manipulating the attention control for the foreground and background videos. Specifically, it proposes a STD-RoPE, an extension of the RoPE, to enhance such an separation. The paper shows good contributions in motivating the proposed problem and designing a corresponding solution to address that, which is appreciated. The paper offers a good experimental analysis to compare with existing works on both training-based and training-free approaches.

**Strengths:**

The motivation of the proposed work sounds reasonable as the paper relays the trajectory control to the separation of foreground and background videos. By analyzing the characteristics of the full attention, it makes sense to achieve such separation via the masking operation in full attention, which can link to individual foreground and background.

The paper also shows a good analysis on why the STD-RoPE was used based on the analysis of attention map.

The paper also shows an extensive experimental comparisons with both training-based and training-free based methods. This demonstrates that the competitive of the proposed method.

**Weaknesses:**

The paper shows several weaknesses that need to be carefully addressed, mainly around several unclear technical details.

1. The principle around STD-RoPE is not very well presented. The paper mentions that the position embedding of video tokens with the bounding box was first selected as the anchor, then position embeddings with all other frames were modified to align with the anchor. If the spatial dimensions are aligned, that seems to contradict to the movement of the trajectory? This part is not well demonstrated in the paper.

2. The Algorithm 1 was not illustrated with details as it was briefly mentioned in STD-RoPE section. It will be nice to offer more details to complement the previous comment on how this STD-RoPE was designed to separate the foreground and background movements associated with the bounding box.

3. When describing the cross-attention in Eqn. (4), it will be nice to provide more clear details on what is the QKV here. This will better explain the injection of the trajectory. It also seems to be not clear on how this trajectory was encoded, especially when the bounding box was used.

4. There are some prior works on point-based trajectory control, it appears the paper does not seem to discuss sufficient relevant works in this direction. What are the main advantage of using bounding box, instead of the point-based control? Can the proposed method still apply to the point-based trajectory control?

5. The paper shows an ablation study in Table 3, where different mechanisms were investigated. It looks like the trajectory control will decrease the video quality. What is the main reason for that?

6. In Fig. 7, the integration of SG yields notable alterations in the video layout. This does not seem to be clear in the results of the given Figure 7. Could you explain what is the notable difference in this layout?

7. By looking at the used instruction template input into the LLM from the Appendix, that was really complex. How does the paper comes with this instruction template and how will others replicate the instruction template?

**Questions:**

1. What are the main advantage of using bounding box, instead of the point-based control? Can the proposed method still apply to the point-based trajectory control?

2. How is the bounding box trajectory was encoded in the cross-attention process?

3. How this STD-RoPE was designed to separate the foreground and background movements associated with the bounding box, especially on the spatially alignment operation.

4. Why will the trajectory control decrease the video quality? Which is the main impacting factor from the proposed designs?

5. How does the paper comes with this instruction template and how will others replicate the instruction template?

---

> ### Author Response · Authors · 2025-11-21
>
> Thank you for your detailed, comprehensive feedback and constructive suggestions. Below are our detailed responses.
>
> ---
>
> Firstly, we clarify some **details regarding the bounding-box trajectory**.
>
> The trajectory $\mathbb{T}$ is composed of *f*-frame bounding-boxes: $\mathbb{T}=\{\mathcal{B}_1,\mathcal{B}_2,\dots ,\mathcal{B}_f\}$. Each bounding-box $\mathcal{B}$ is defined by the relative position coordinates of its top-left and bottom-right corners. Thus, during the generation process, we can determine which video tokens are within the trajectory region. These tokens are the ones we want to generate as the foreground, while the rest are background tokens.
>
> ---
> > **W1: if the spatial dimensions are aligned, that seems to contradict to the movement of the trajectory**
>
> Thank you for your question. We only modify the position embedding of the video tokens within the trajectory region, **without changing their order of arrangement**. Through the alignment of their spatial dimensions, we enhance inter-frame attention among foreground tokens, facilitating the generation of more relevant content in the foreground regions across different frames, thereby ensuring that the trajectoriy of object in the generated video is consistent with the target trajectoriy.
>
> ---
> > **W2&Q3: lack of details regarding STD-RoPE**
>
> We assume that the k-th frame is selected. We can identify the anchor-tokens based on $\mathcal{B}_k$ and get their position embedding $PE[k,\mathcal{B}\_k]$, i.e., the anchor. Subsequently, we align the spatial dimensions of the position embedding of the foreground region tokens in all other frames with it:
> $$
> PE\_{spatial}[i, \mathcal{B}\_i] ← PE\_{spatial}[k, \mathcal{B}\_k],i \in \{1,...,k-1,k+1,...,f\}
> $$
>
> ---
> > **W3&Q2: lack of details regarding cross-attention**
>
> - **Details about QKV.** The QKV in Eq.4 represent the parameter matrices $W_q$, $W_k$, and $W_v$, which are used to calculate the query, key, and value in the attention computation, respectively. In Wan2.1, the query is computed from the video tokens, while the key and value are derived from the text conditions.
>
> - **Details about bounding-box.** During the cross-attention computation, the video tokens are flattened into a sequence. Based on the trajectory $\mathbb{T}$, we can classify the video tokens into foreground tokens and background tokens. With this classification, we can construct a cross-attention mask to ensure that tokens from different regions are guided by their corresponding text conditions.
>
> ---
> > **W4&Q1: point-based method**
>
> Thank you for your question. We do not extensively discuss point-based methods because the objectives of these two trajectory control methods are inconsistent. The bounding-box-based method focuses on the overall trajectory of the object (e.g., the entire dog's body), while the point-based method pays more attention to the local movement trajectory (e.g., the dog's nose).
>
> Compared with point-based methods, the main advantage of the bounding-box-based method is that it enables the control of both object's size and its overall movement. By treating the center of the bounding obx as the point, our method can be roughly considered a point-based method.
>
> ---
> > **W5&Q4: the trajectory control decrease the video quality**
>
> In fact, compared with original model, the introduction of the SG module leads to an improvement in video quality. After further incorporating STD-RoPE, there is a slight decline in video quality. We attribute this to the modification of position embedding by STD-RoPE, which causes a shift in the model's output distribution. Overall, **the video quality generated by DiTraj outperforms that of original model**.
>
> ---
> > **W6: notable difference in video layout in Fig.7**
>
> The first column in Fig.7 shows the output of the original model, which does not conform to the target layout at all. In the second column (w/ SG), part of the horse's body extends beyond the bounding box range, while in the third column (DiTraj), the horse's body is almost entirely within the bounding box.
>
> ---
>
> > **W7&Q5: instruction template**
>
> Tanks for your question. We simply list several rules and requirements that the LLM needs to follow. The construction of our template is heuristic, aiming to present our requirements to the LLM as completely as possible. In our method, this template can be used directly without any modification. If the LLM is required to complete other tasks, similar templates can be constructed heuristically, and the requirements should be stated as completely as possible. The construction of instruction templates does not need to be overly rigid, as the latest LLMs currently possess sufficient comprehension capabilities and strong adherence to instructions.
>
> ---
>
> All the additional details mentioned above will be included in the revised version of the paper.
>
> We are grateful again for your insightful comments. Please let us know if any further elaboration is needed.

---

> ### Author Response · Authors · 2025-11-27
>
> Dear Reviewer ddfm,
>
> We would like to kindly check if our previous response clarifies your questions. We are happy to provide any further details if needed.
>
> Thank you for your time and effort in reviewing our paper.
>
> Best regards,
>
> The Authors

---

> ### Comment · Reviewer_ddfm · 2025-11-27
>
> Thanks for your rebuttal. I think my concerns have been addressed now. The instruction template is still something that should be considered to improve in the future on how these rules and requirements can be automatically formulated without much handcrafted designs. This could further help strengthen this work.

---

> > ### Author Response · Authors · 2025-11-28
> >
> > Dear reviewer ddfm,
> >
> > Thank you for reviewing our paper and providing valuable feedback. We're glad our rebuttal addressed your concerns. We will endeavor to refine the instruction template according to your recommendations.
> >
> > Thanks again for the time and effort in reviewing our work.
> >
> > — Authors

---

### Official Review · Reviewer_Kuu7 · 2025-10-31

**Soundness:** 2
**Presentation:** 3
**Contribution:** 3
**Rating:** 4
**Confidence:** 5

**Summary:**

The paper proposes a novel framework for precise object trajectory control in text-to-video (T2V) generation, specifically designed for Diffusion Transformer (DiT)-based models. The authors leverages an LLM to decompose user prompts into foreground (object) and background descriptions, which are used to guide video generation via a cross-attention mask based on bounding-box trajectories. A method to modify position embeddings of foreground tokens across frames is proposed, aligning spatial dimensions to enhance cross-frame attention while preserving temporal coherence, thereby improving trajectory control. This is complemented by an R-token mask to avoid artifacts.

**Strengths:**

1. This paper innovatively combines prompt decomposition based on large language models with positional embedding operations in deep image processing, offering novelty compared to methods focused on U-Net. STD-RoPE represents an innovative application of 3D-RoPE to trajectory control.
2. The methodology employs a step-by-step exposition supplemented by illustrative diagrams to enhance comprehension. The writing is precise and structurally rigorous.

**Weaknesses:**

1.The foreground-background separation relies on an external large language model (Qwen3). If the model misinterprets the prompt, errors may be introduced.
2. This method requires predefined trajectories. Some metrics set in the paper are not reasonable, and it fails to demonstrate whether this approach leads to reduced video quality. Video quality metrics should be added (such as fvd). Compared to prior work, the visualized scenes do not exhibit complex motion trajectories. Although the paper mentions optimization for large trajectories, the actual visualizations do not show significantly complex or varied trajectory movements.
3.Experiments focus on single-object scenarios with simple backgrounds. Performance in cluttered or multi-object environments remains unverified.
4.Text in images appears overly blurred and too small. Font formatting lacks consistency, resulting in an unprofessional appearance.

**Questions:**

1. How does the method handle cases where the LLM generates inaccurate foreground/background prompts? Could a rule-based fallback be integrated to reduce dependency? How exactly is the LLM used for separation? What is the predefined prompt? Does it need modification?
2. Is there a plan to extend DiTraj for interactive trajectory editing during generation, rather than relying on pre-specified boxes?
3. The paper tests on Wan2.1 and CogVideoX. How portable is DiTraj to other DiT-based models (e.g., Sora-inspired architectures)?Could the performance improvement stem from changes in the video model?

---

> ### Author Response · Authors · 2025-11-21
>
> We sincerely thank the reviewer for the insightful comments.
>
> ---
>
> > **W1&Q1: errors introduced by large language models**
>
> Thank you for pointing out the potential errors that may be introduced by LLM. We provide our response below:
>
> Our template contains theresponse rules and our requirements(please see the Appendix A.2.3). Users can embed their prompts into the template and input them to the LLM without any modifications.
>
> To verify the capabilities of the LLM, we collect 120 prompts for foreground-background separation, which cover complex scenarios and ambiguous expressions (e.g., "A Red Fox Navigating the Winter Wetland Maze.", "A bird on a car." etc.). These prompts are combined with our template and input into the LLM to obtain the results, which are then evaluated by humans to determine whether they meet expectations.
>
> |model| success | error | acc |
> |:-----|:------|:------|:------|
> |GPT-4o| 119 | 1 | 99.17%|
> |Qwen3-7B| 117 | 3 | 97.50%|
>
> The experimental results demonstrate that the LLM rarely makes mistakes and exhibits high reliability.
>
> ---
>
> > **W2: requires predefined trajectories**
>
> The trajectories are fully scalable and can be entirely constructed by the users themselves, rather than being limited to a fixed set of trajectories. The manually constructed trajectories and the prompt set that we provided are solely for the purpose of testing the effectiveness of our method and comparing it with previous methods. We will include interactive trajectory construction tools in the released code.
>
> ---
>
> > **W2: metrics in quantitative comparison**
>
> For the video quality, we evaluate the metrics on Vbench. As shown in **Table 3**, our method does not degrade video quality.
>
> Additionally, following your suggestion. We have supplemented the FVD, FID, and IS metrics. We use 500 randomly selected videos from AnimalKingdom[1] as the real distribution. The results are as follows:
>
> |method|FVD↓|FID↓|IS↑|
> |:-----|:------|:------|:------|
> |FreeTraj|1946|101.4|14.78|
> |Peekaboo|1287|90.64|13.08|
> |Trailblazer|1336|89.32|15.25|
> |Direct-a-Video|1455|102.8|13.73|
> |LVD|1288|99.81|14.19|
> |Tora|1198|91.33|15.79|
> |**DiTraj**|**1168**|**89.08**|**15.91**|
>
> We further compared DiTraj against the base model, and the results show that while FID exhibits a minor drop, both FVD and IS are significantly enhanced:
>
> |method|FVD↓|FID↓|IS↑|
> |:-----|:------|:------|:------|
> |Wan2.1|1412|**87.64**|15.88|
> |Wan2.1 + **DiTraj**|**1168**|89.08|**15.91**|
>
> ---
>
> > **W2&W3: simple provided visualization examples**
>
> Thank you for the valuable feedback. Due to space limitations, we were unable to present more cases in the manuscript. We sincerely hope that you can refer to the **supplementary materials** and **Appendix A.4** for additional cases involving more complex scenarios and trajectories. Moreover, following your suggestion, we will replace the relatively simple cases in the qualitative comparison with more complex ones in the revised version.
>
> ---
>
> > **W3: multi-object scenarios**
>
> Thank you for the insightful questions. DiTraj is the first training-free framework tailored specifically for DiT for **single-object** trajectory controllable video generation. Multi-object trajectory control is a more challenging research direction that requires simultaneously considering the trajectories and interactions of multiple objects. The foreground-background separation guidance of DiTraj can be extended to scenarios involving multiple objects. Moreover, STD-RoPE, by merely modifying the position embedding, enhances the trajectory control capability for single objects, which can offer some insights for multi-object scenarios. Therefore, as a promising feasible approach, we will open-source all code of this work to facilitate the collective advancement of research on multi-object trajectory control technologies.
>
> ---
>
> > **W4: the issue of text in the images**
>
> Thank you for your comprehensive and detailed reply. We have fixed these issues in the revised version.
>
> ---
>
> > **Q2: extend DiTraj for interactive trajectory editing during generation, rather than relying on pre-specified boxes**
>
> Thank you for pointing out a extensible research direction for our next step. But it should be noted that in our work:
>
> First, **in the user input phase**, trajectories are fully scalable and can be entirely interactively constructed by the users themselves, rather than being limited to a fixed set of trajectories.
>
> Second, **in diffusion phase**, the trajectory of the object is typically generated as a whole. In short, for the task of editing trajectories while maintaining consistent video content, we can design memory modules and combine them with inversion methods for further exploration. We plan to research this in the future.
>
> ---
>
>
> References:
>
> [1] Animal kingdom: A large and diverse dataset for animal behavior understanding. CVPR2022

---

> ### Author Response · Authors · 2025-11-21
>
> > **Q3: other DiT-based model**
>
> Our method can be extended to the 3D full attention DiT video generation model. In addition to Wan2.1 and CogVideoX, we have also adapted our method to HunyuanVideo and Open-Sora 2.0. The quantitative experimental results are as follows:
>
> |base model w/ DiTraj|SC↑|BC↑|MS↑|AQ↑|IQ↑|Cov↑|mIoU↑|CD↓|AP50↑|
> |:-----|:------|:------|:------|:------|:------|:------|:------|:------|:------|
> |Wan2.1-1.3b|0.937|0.957|0.990|0.627|0.677|0.96|47.3|0.09|50.5|
> |Wan2.1-14b|**0.941**|0.960|**0.991**|**0.633**|**0.683**|**0.96**|47.2|**0.09**|51.7|
> |CogvideoX-5b|0.935|0.956|0.990|0.580|0.652|0.94|45.2|0.14|**58.8**|
> |HunyuanVideo|0.931|0.955|0.984|0.617|0.661|0.95|47.8|0.11|49.6|
> |OpenSora2.0|0.934|**0.961**|0.978|0.626|0.658|0.95|**48.2**|0.10|52.2
>
> The results indicate that the quality of the generated videos improves as the base model's generation capabilities are enhanced.
>
> ---
> We are grateful again for your insightful comments. Please let us know if any further elaboration is needed.

---

> ### Author Response · Authors · 2025-11-27
>
> Dear Reviewer Kuu7,
>
> We would like to kindly check if our previous response clarifies your questions. We are happy to provide any further details if needed.
>
> Thank you for your time and effort in reviewing our paper.
>
> Best regards,
>
> The Authors

---

### Official Review · Reviewer_ZZxd · 2025-11-03

**Soundness:** 2
**Presentation:** 3
**Contribution:** 3
**Rating:** 4
**Confidence:** 4

**Summary:**

In this paper, the authors focus on achieving training-free trajectory control for Diffusion Transformer (DiT)-based video generation models through the control condition of bounding boxes. To this end, the authors propose the framework, DiTraj, which primarily consists of two key modules: foreground-background separation guidance and inter-frame Spatial Temporal Decoupled 3D-RoPE (STD-RoPE). Specifically, the model is guided to focus on the foreground and background regions. Furthermore, by observing and modifying the 3D full attention and 3D-RoPE mechanisms, the authors achieve trajectory control of the object. Extensive experiments demonstrate the effectiveness of the proposed framework.

**Strengths:**

1. The issue focused on by the paper is important. How to achieve user-friendly and efficient control for video generation models, including DiT-based models, is a topic of widespread interest.

2. The analysis of the underlying mechanism is clear. The paper conducts a detailed analysis of the 3D full attention map and 3D-RoPE within the DiT architecture and proposes a specific method for training-free trajectory control.

**Weaknesses:**

1. [Major] Limited diversity of experiments to demonstrate the robustness of the framework: In the qualitative comparison, the provided visualization examples are primarily limited to simple scenarios with a single foreground object undergoing significant motion. This raises concerns about the method’s applicability and robustness in several other key scenarios, such as cases with multiple foreground objects, situations involving small-magnitude object motion, or more complex scenes.

2. [Major] Potential limited scope of method’s applicability: The proposed method achieves trajectory control by modifying the 3D-RoPE of tokens. However, this approach may perform very poorly or be nearly unusable in certain scenarios. For example, consider a case where multiple initially interacting foreground objects are intended to gradually separate (e.g., two penguins hugging each other and then moving in different directions). This is because when foreground objects have complex interactions, it may become extremely difficult to accurately disentangle and modify the 3D-RoPE for tokens corresponding to each individual object.

3. [Minor] The paper may overclaim the effectiveness of its ‘3D-aware’ trajectory control. The demonstrated ‘3D-aware’ control is primarily limited to cases of object scaling (zooming in and out), and does not encompass all the 3D motion cases. For instance, it is unclear whether the method can handle a common 3D scenario, such as making a foreground vehicle follow a specific trajectory to drive in a circle to show the vehicle in different directions.

**Questions:**

1. Regarding the robustness and applicability of the framework (related to Weaknesses 1 & 2):

(1) On multi-object scenarios: The experiments primarily show single-object trajectory control. Could the authors elaborate on how the proposed framework, particularly the M^cross mask and STD-RoPE modification, would theoretically handle scenarios with multiple foreground objects? It is recommended to include a discussion or a qualitative example of a multi-object scene in their rebuttal.

(2) On handling complex interactions: Another key challenge arises when objects interact physically, such as the example of “two penguins hugging each other and then separating.” Could the authors explain how to address instances where the bounding boxes overlap or interact? It is recommended to include a discussion and some qualitative examples of complex interactions in their rebuttal.

2. Regarding the claims of ‘3D-aware’ control (related to Weakness 3):

The term ‘3D-aware’ may suggest an understanding of object geometry and viewpoint. The current method demonstrates excellent control over object scaling to simulate depth. To clarify the scope of this claim, could the authors comment on whether their method can handle viewpoint changes that require rendering an object from different viewpoints? For example, could it generate a car driving in a circle while showing its front, side, and rear views accordingly? Perhaps rephrasing the contributions as  ‘2.5D’ or ‘scale control’ would be more appropriate.

---

> ### Author Response · Authors · 2025-11-21
>
> We sincerely thank the reviewer for the thoughtful questions and constructive feedback. Below are our detailed responses.
>
> ---
>
> > **W1: simple provided visualization examples**
>
> Thank you for the valuable feedback. Due to space limitations, we were unable to present more cases in the manuscript. We sincerely hope that you can refer to the **supplementary materials** and **Appendix A.4** for additional cases involving more complex scenarios and trajectories. Moreover, following your suggestion, we will replace the relatively simple cases in the qualitative comparison with more complex ones in the revised version.
>
> ---
>
> > **W1&W2&Q1: multi-object scenarios and complex interactions**
>
> Thank you for the insightful questions. DiTraj is the first training-free framework tailored specifically for DiT for **single-object** trajectory controllable video generation. Multi-object trajectory control is a more challenging research direction that requires simultaneously considering the trajectories and interactions of multiple objects. The foreground-background separation guidance of DiTraj can be extended to scenarios involving multiple objects. Moreover, STD-RoPE, by merely modifying the position embedding, enhances the trajectory control capability for single objects, which can offer some insights for multi-object scenarios. Therefore, as a promising feasible approach, we will open-source all code of this work to facilitate the collective advancement of research on multi-object trajectory control technologies.
>
> ---
>
> > **Q2: the claims of ‘3D-aware’ control**
>
> Thank you for your insightful comments. Considering that our work indeed lacks control over the viewpoint of object, thus we accept your suggestion, and change the **3D-aware** to **2.5D-aware** in the revised version.
>
> ---
> We are grateful again for your insightful comments and believe the above clarifications address your key concerns. Please let us know if any further elaboration is needed.

---

> ### Author Response · Authors · 2025-11-27
>
> Dear Reviewer ZZxd,
>
> We would like to kindly check if our previous response clarifies your questions. We are happy to provide any further details if needed.
>
> Thank you for your time and effort in reviewing our paper.
>
>
> Best regards,
>
> The Authors

---

### Meta-Review · Area_Chair_JtN1 · 2026-01-03

**Summary:**

The paper presents a method (DiTraj) for trajectory-controlled video generation using a DiT framework with a novel STD-RoPE mechanism and LLM-guided foreground-background separation. While the idea of integrating explicit trajectory control is valuable and the reported results show promise, the manuscript in its current form is ‌not acceptable for publication‌. It suffers from significant issues regarding ‌experimental validation, technical clarity, and methodological justification‌, which severely undermine its claims and contributions.

‌**1. Insufficient and Narrow Experimental Validation:‌**

‌**Lack of Robustness Demostration‌:** A consensus issue (R1, R2, R3). The experiments are narrowly focused on simple, single-object motions against clean backgrounds. The paper fails to demonstrate performance in the complex, cluttered, and multi-object scenarios that are crucial for proving real-world applicability and robustness. Claims about handling "large trajectories" are not visually substantiated (R2).

‌**Overclaimed Capabilities‌:** The paper's title and claims suggest "3D-aware" trajectory control, but the evidence is limited mainly to 2D planar motion and zooming. There is no proof of handling true 3D motion (e.g., object rotation, following non-planar paths) as pointed out by R1.

‌**Incomplete Evaluation‌:** The absence of standard video quality metrics like Fréchet Video Distance (FVD) is a serious omission (R2). Ablation studies are surface-level and lack analysis (e.g., why does trajectory control lower quality in Table 3? (R3)).

‌**2. Poor Technical Clarity and Explanation:‌**

‌**Unclear Core Mechanism‌:** The operation and justification of the key ‌STD-RoPE‌ component are poorly explained (R3, R4). Reviewers find the description contradictory (aligning spatial embeddings seems to negate movement) and lack necessary details (R3). The semantic meaning of the modified position embeddings is never clarified, raising fundamental concerns about what the model is actually learning (R4).

‌**Vague Trajectory Injection‌:** The process of injecting trajectory information via cross-attention (Eq. 4) is not described with sufficient technical detail regarding the QKV operations and bounding box encoding (R3).

‌**Opaque and Complex LLM Pipeline‌:** The heavy reliance on a complex, pre-defined LLM prompt for foreground separation is a potential point of failure. The paper does not discuss robustness to LLM errors (R2, R4) or explain the genesis and reproducibility of the complex instruction template (R3).

‌**3. Methodological and Presentation Issues:‌**

‌**Novelty and Justification Gap‌:** R4 highlights that the core control mechanism builds on well-established practices, and the specific design choices within STD-RoPE (e.g., preserving temporal dimensions) lack theoretical or principled justification, appearing ad-hoc.

**‌Inadequate Related Work‌:** The discussion fails to properly situate the method against prior art in point-based trajectory control, missing an opportunity to justify the choice of bounding boxes and define the contribution boundary (R3).

**‌Unprofessional Presentation‌:** Issues with figure quality, text formatting, and labeling (e.g., potential mislabeling in tables) detract from the manuscript's credibility (R2, R3).

Overall, this paper requires substantial work before it can be reconsidered. Only after addressing these foundational issues can the paper's true contribution—a potentially useful but currently undersupported and unclear method for trajectory control—be fairly evaluated.

**Reviewer Concerns:**

1. The main concerns on multi-object scenarios and handling complex interactions of R1 and R2 were not addressed by the rebuttal. The added examples in A.4 are still simple, and how the propsoed mehtod can be extended to these complex cases are not clarified.

2. The main concerns of R3 and R4 were addressed by the rebuttal.

**Reviewer Scores:**

Note that both R1 and R2 gave negative scores with high confidence. Since their main concerns were not addressed by the rebuttal, I tend to recommend a rejcet for this paper.

---

### Decision · Program_Chairs · 2026-01-26

Reject